# Resource- and Neighbor-Aware Observation Transmission Scheme in Satellite Networks

**DOI:** 10.3390/s23104889

**Published:** 2023-05-19

**Authors:** Haneul Ko, Yeunwoong Kyung

**Affiliations:** 1Department of Electronic Engineering, Kyung Hee University, Yongin-si 17104, Republic of Korea; heko@khu.ac.kr; 2Division of Information & Communication Engineering, Kongju National University, Cheonan-si 31080, Republic of Korea

**Keywords:** satellite, observation, stochastic game, optimization, delay

## Abstract

The observation satellite can exploit its own storage and computational resources to reduce the transmission delay. However, too excessive usage of these resources can have negative effects on the queuing delay at the relay satellite and/or on conducting other tasks at each observation satellite. In this paper, we proposed a new resource- and neighbor-aware observation transmission scheme (RNA-OTS). In RNA-OTS, each observation satellite decides whether to use its resources and the resources of the relay satellite at each time epoch by considering its resource utilization and transmission policies of neighbor observation satellites. For the optimal decision of each observation satellite in a distributed manner, the operation of observation satellites is modeled by means of a constrained stochastic game, and a best-response-dynamics-based algorithm is devised to find the Nash equilibrium. The evaluation results demonstrate that RNA-OTS can decrease the delay to deliver the observation to the destination by up to 87% compared to a relay-satellite-based scheme while guaranteeing a sufficiently low average utilization of the resources of the observation satellite.

## 1. Introduction

Observation satellites are commonly used to watch particular targets for a variety of tasks, such as environmental monitoring and catastrophe surveillance due to their overflight capability [1,2]. However, wireless resources and the transmission power of observation satellites are limited, and thus sufficiently high transmission throughput cannot be achieved. Moreover, the opportunities to transmit the observations to the ground station are quite restricted [3]. That is, as shown in Figure 1, even though the observation satellite observes the target, it cannot immediately transmit the observation. It is only after the observation satellite contacts the ground station that it can transmit the observation to the ground station. As a real example, within a system period of around 100 min, a typical observation satellite may contact a specific ground station in less than 10 min [4]. These limitations cause the increased delay in delivering the observation to the destination. To mitigate this problem, we can exploit the resources of two types of satellites: (1) resources of the relay satellite; and (2) resources (i.e., storage and computational resources) of the observation satellite. When using the resources of the relay satellite, the observation satellites can forward their observations to the relay satellite. Then, the relay satellite can relay them to the destination. However, when lots of observation satellites simultaneously forward the observations to the relay satellite, the delay can increase due to the queuing in the relay satellite. To avoid the excessive simultaneous transmissions, each observation satellite does not forward the observation when lots of neighbor observation satellites are forwarding the observations to the relay satellite. Instead of that, the observation satellite can store its observation (i.e., exploit the storage resources of the observation satellite) until the connection to the ground station becomes available. Then, the observation satellite can compress the observation to reduce the transmission delay by using its computational resources. However, when using excessive storage and computational resources, the observation satellite cannot store and conduct other tasks (e.g., image processing for environmental monitoring) in a timely manner. To sum up, the resources of two types of satellites should be appropriately utilized. Even though a centralized approach may aid in coordinating the utilization of two types of resources, a significant signaling overhead can arise and a centralized entity can become overburdened owing to a high computation overhead.

In this paper, we propose a resource- and neighbor-aware observation transmission scheme (RNA-OTS). In RNA-OTS, to avoid the excessive simultaneous transmissions to the relay satellite (i.e., to avoid the increased queuing delay at the relay satellite), each observation satellite decides whether or not to transmit its observation to the relay satellite (or to the ground station) (i.e., whether or not to use the resources of the relay satellite) by considering the neighbor observation satellites’ transmission policies. Additionally, to conduct other tasks (e.g., image processing for environmental monitoring) in a timely manner, each observation satellite decides whether or not to delay the transmission and to compress the observation before the transmission (i.e., whether or not to use its own resources) with a consideration of its current resource utilization. For the distributed optimal decision of each observation satellite by considering the transmission policies of neighbor observation satellites and its resource utilization, the operation of observation satellites is modeled by means of a constrained stochastic game. Then, a best-response-dynamics-based algorithm is devised to find the Nash equilibrium of the game. The Nash equilibrium indicates the status where the best performance (i.e., minimized delay) can be achieved and no players can profit from arbitrarily deviating from the equilibrium [5,6]. The evaluation results demonstrate that RNA-OTS can decrease the delay to deliver the observation to the destination by up to 87% compared to a relay-satellite-based scheme while guaranteeing a sufficiently low average utilization of the resources of the observation satellite. Moreover, it can be found that RNA-OTS can adaptively operate according to its operating environments.

The contributions of this paper can be summarized as follows: (1) by means of a constrained stochastic game model, the transmission policy of each observation satellite can be optimized by considering the other observation satellites’ transmission policies in a distributed manner; (2) since the optimal transmission policy can be attained with little overhead, the proposed scheme can readily achieve the best performance in real systems; and (3) we evaluate RNA-OTS in a variety of situations, and the results may be used as design guides for observation satellite systems.

The remainder of this paper is constructed as follows. First, we describe related works in Section 2 and elaborate RNA-OTS in Section 3. The constrained stochastic game model is defined in Section 4. Then, the evaluation results are given in Section 5. The conclusion is presented in Section 6.

## 2. Related Work

To support the transmission of the observation satellite, many works have been conducted [7,8,9,10,11,12,13,14,15,16,17]. These can be categorized into (1) non-collaborative approach (i.e., without the relay satellite) [7,8,9,10,11,12]; and (2) collaborative approach (i.e., with the relay satellite) [13,14,15,16,17]. Note that our work in this paper can be categorized as a collaborative approach.

Gooley et al. [7] firstly introduced a satellite range scheduling problem to schedule the satellite transmissions with the ground stations within the contact time between them. A mixed-integer programming (MIP) problem is formulated to maximize the total number of satellite transmissions with ground stations and solved by a heuristic method. Meng-Gérard et al. [8] presented a problem for scheduling satellite tasks to maximize the profit by processing the tasks. Marinelli et al. [9] formulated a multiprocessor task scheduling problem to maximize the number of satellite missions with ground stations and developed a Lagrangian-based heuristic method to solve the problem. Spangelo et al. [10] provided a MIP model and algorithm for the communication scheduling problem considering single-satellite and multi-ground stations to maximize the total amount of transmitted data. Cho et al. [11] proposed a binary integer linear programming formulation for the satellite task scheduling to maximize the overall mission performance metric. Lin et al. [12] formulated a non-convex optimization problem to maximize the energy efficiency of satellite-terrestrial integrated networks and proposed two beam-forming schemes with low complexity to solve the formulated problem. Lin et al. [18] formulated an optimization problem to maximize the secrecy energy efficiency while maintaining the total transmission power below a certain level, and decomposed the original optimization problem into sub-problems to obtain the near-optimal solution.

Note that the above-mentioned previous works focused on the direct transmission between observation satellites and ground stations; however, the opportunities for this direct transmission are limited due to the short connection time between observation satellites and ground stations. To alleviate this problem, we can introduce relay satellites that always have connection to ground stations.

Jia et al. [13] provided a collaborative data download scheme to schedule communications links both between satellites and between satellite and ground stations to maximize the throughput. Zhou et al. [14] presented a dynamic programming framework based on the Markov decision process (MDP) to optimize the transmission scheduling by considering the stochastic data arrival characteristics, the inter-satellite contact model, and the energy dynamics model. Deng et al. [15] proposed a two-phase task scheduling method to dynamically adjust the scheduling according to the frequent task variation in relay-satellite-based systems. Wu et al. [16] introduced a priority-based task scheduling algorithm considering the different requirements of emergency and normal tasks. Chen et al. [17] proposed a data breakpoint transmission method which can divide a single task into multiple subtasks and schedule them in multiple contact time periods to maximize the task completion rate.

Table 1 summarizes the comparison of related collaborative approaches. Even though these works focused on the transmission scheduling or task scheduling with the relay satellites (i.e., how to use the resources of the relay satellites), it was not investigated how to utilize the storage and computational resources of the observation satellites.

## 3. Resource- and Neighbor-Aware Observation Transmission Scheme in Satellite Networks

Figure 2 shows the system model in this paper. There are *N* observation satellites. These observation satellites can observe the target when the target is in their coverage. After the observation, each observation satellite needs to deliver the observation to the destination at the ground. For this, the observation satellite can directly transmit the observation to the ground station; however, the observation satellite can transmit the observation to the ground station only when it can have a connection with the ground station, and the opportunities for the connection with the ground station are quite limited [4]. Moreover, even if the observation satellite has a connection with the ground station, the transmission power of the observation satellite is limited. Therefore, a large delay to deliver the observation to the destination can be expected.

To reduce the delay to deliver the observation to the destination, RNA-OTS periodically decides whether to exploit the resources of relay and observation satellites. Specifically, regarding the usage of the resources of the relay satellite, each observation satellite periodically decides whether to transmit its observation to the relay satellite (or to the ground station) (i.e., whether to use the resources of the relay satellite). When the observation satellite decides to use the resources of the relay satellite, it forwards the observation to the relay satellite at geostationary orbit that always has connection to the ground station. Then, the relay satellite can relay them to the destination. However, when lots of observation satellites simultaneously forward the observations to the relay satellite (i.e., when excessively using the resource of the relay satellite), the queuing delay at the relay satellite can increase. Therefore, each observation satellite should consider neighbor observation satellites’ actions and decide whether to forward its observation to the relay satellite.

On the other hand, regarding the usage of the resources of the observation satellite, each observation satellite periodically decides whether or not to delay the transmission and to compress the observation before the transmission (i.e., whether to use the storage and computational resources of the observation satellite). However, when excessively using the resources of the observation satellites, the observation satellite cannot conduct its own important tasks (e.g., image processing for environmental monitoring) in a timely manner.

To sum up, it should be appropriately decided whether to use the resources of two types of satellites by considering their operating environments. Especially for the decision on the usage of the resources of the relay satellite, each observation satellite should consider the transmission policies of neighbor observation satellites to avoid the increased queuing delay at the relay satellite. In addition, for the decision on the usage of its own resources, each observation satellite should consider the average utilization of its resources. For the distributed optimal decision of each observation satellite by considering the transmission policies of neighbor observation satellites and their resource utilization, a constrained stochastic game is developed in the next section.

## 4. Constrained Stochastic Game

When using appropriately the resources of two types of satellites by considering the operating environments, the delay to deliver the observation to the destination can be minimized and the average utilization of resources of the observation satellite can be kept below certain levels, which are the objectives of the formulated constrained stochastic game model [19]. In our model, observation satellites are the players. Thus, observation satellite *i* and player *i* are interchangeable in the formulation. The important notations are summarized in Table 2 (in this paper, the bold notation indicates the space).

### 4.1. State Space

Let S = $\prod_{i}S_{i}$ denote a global state space of players (i.e., observation satellites), where $S_{i}$ is a local state space of player *i* and ∏ means the Cartesian product. $S_{i}$ can be defined as
(1)$$\begin{array}{c} S_{i}=M_{i}\times G_{i} \end{array}$$
where $M_{i}$ and $G_{i}$ denote the observation state space of observation satellite *i* and the ground station state space of observation satellite *i*, respectively.

Since $M_{i}$ can represent whether observation satellite *i* observes a target or not and the elapsed time from the observation, $M_{i}$ can be represented as
(2)$$\begin{array}{c} M_{i}=\left\{0,1,2,\ldots ,M_{\max}\right\} \end{array}$$
where $M_{i}\left(\in M_{i}\right)=0$ denotes that observation satellite *i* does not observe any target. In addition, $M_{i}$ (>0) represents that observation satellite *i* observes a target and the elapsed time from the observation is $M_{i}$. $M_{\max}$ is the maximum elapsed time in our system model.

Meanwhile, $G_{i}$ can be defined as
(3)$$\begin{array}{c} G_{i}=\left\{0,1\right\} \end{array}$$
where $G_{i}\left(\in G_{i}\right)$ denotes whether observation satellite *i* can connect with the ground station or not. That is, if $G_{i}=0$, observation satellite *i* cannot connect with the ground station. Otherwise, it can connect with the ground station.

### 4.2. Action Space

Let A = $\prod_{i}A_{i}$ denote a global action space, where $A_{i}$ denotes a local action space of player *i*. $A_{i}$ can be represented as follows. (Note that the two types of decisions (i.e., the decision on whether to use the resource of the relay satellite and the decision on whether to use the resources of the observation satellite) can be considered by the defined one-dimensional action space).
(4)$$\begin{array}{c} A_{i}=\left\{0,1,2,3,4\right\} \end{array}$$
where $A_{i}\left(\in A_{i}\right)=0$ represents the situation where observation satellite *i* delays its transmission. In addition, $A_{i}=1$ and $A_{i}=2$ denote the situation where observation satellite *i* transmits its observation to the relay satellite and ground station without the compression, respectively. Moreover, $A_{i}=3$ and $A_{i}=4$ denote the situation where observation satellite *i* transmits its observation to the relay satellite and ground station after the compression, respectively (any kind of compression method can be exploited). Note that $A_{i}=2$ and $A_{i}=4$ can be conducted only when observation satellite *i* has a connection with the ground station (i.e., $G_{i}=1$).

### 4.3. Transition Probability

The transition probability $P[S{}_{i}^{\prime }|S_{i},A_{i}]$ means the probability of transiting to the next state $S_{i}^{\prime }$, given the current state $S_{i}$ and action $A_{i}$ of observation satellite *i*. The state $M_{i}$ is affected by the chosen action (e.g., when observation satellite *i* transmits its observation, $M_{i}$ becomes 0), whereas the state $G_{i}$ is not influenced by the chosen action. In addition, these states change independently of each other. Therefore, $P[S_{i}^{\prime }|S_{i},A_{i}]$ between the current state $S_{i}=\left[M_{i},G_{i}\right]$ and the next state $S_{i}^{\prime }=\left[M_{i}^{\prime },G_{i}^{\prime }\right]$ can be described as
(5)$$P\left[S_{i}^{\prime }|S_{i},A_{i}\right]=P\left[M_{i}^{\prime }|M_{i},A_{i}\right]\times P\left[G_{i}^{\prime }|G_{i}\right].$$

When it is assumed that the inter-observation time of observation satellite *i* follows the exponential distribution and its mean is $1/\lambda_{i}^{M}$, the probability that observation satellite *i* observes the target during the decision epoch τ can be calculated as $\lambda_{i}^{M}\tau$ [20,21]. Thus, $P[M_{i}^{\prime }|M_{i}=0,A_{i}]$ can be represented as
(6)$$\begin{array}{c} P\left[M{}_{i}^{\prime }|M_{i}=0,A_{i}\right]=\left\{\begin{array}{c} 1-\lambda_{i}^{M}\tau ,\;\;\mathrm{if}\;M{}_{i}^{\prime }=0\\ \lambda_{i}^{M}\tau ,\;\;\;\;\;\;\;\;\;\;\mathrm{if}\;M{}_{i}^{\prime }=1\\ 0,\;\;\;\;\;\;\;\;\;\;\;\;\;\;\mathrm{otherwise}. \end{array}\right. \end{array}$$

Meanwhile, if observation satellite *i* does not transmit its observation (i.e., $A_{i}=0$), the elapsed time from the observation increases one by one until $M_{i}$ reaches the maximum elapsed time. Therefore, the corresponding transition probabilities can be described as
(7)$$\begin{array}{c} P\left[M{}_{i}^{\prime }|0<M_{i}<M_{\max},A_{i}=0\right]=\left\{\begin{array}{c} 1,\;\mathrm{if}\;M{}_{i}^{\prime }=M_{i}+1\\ 0,\;\mathrm{otherwise} \end{array}\right. \end{array}$$
and
(8)$$\begin{array}{c} P\left[M{}_{i}^{\prime }|M_{i}=M_{\max},A_{i}=0\right]=\left\{\begin{array}{c} 1,\;\mathrm{if}\;M{}_{i}^{\prime }=M_{i}\\ 0,\;\mathrm{otherwise}. \end{array}\right. \end{array}$$

When observation satellite *i* transmits its observation (i.e., $A_{i}\neq 0$), $M_{i}$ will be always 0. Therefore, $P[M{}_{i}^{\prime }|M_{i}\neq 0,A_{i}\neq 0]$ can be represented as
(9)$$\begin{array}{c} P\left[M{}_{i}^{\prime }|M_{i}\neq 0,A_{i}\neq 0\right]=\left\{\begin{array}{c} 1,\;\mathrm{if}\;M{}_{i}^{\prime }=0\\ 0,\;\mathrm{otherwise}. \end{array}\right. \end{array}$$

We assume that the duration of the disconnection (connection) between the ground station and observation satellite *i* follows the exponential distribution with mean $1/\lambda_{i}^{D}$ ($1/\lambda_{i}^{C}$). Thus, the probability that observation satellite *i* connects with the ground station during τ is $\lambda_{i}^{C}\tau$ [20,21]. In addition, the probability that observation satellite *i* loses the connection with the ground station during τ is $\lambda_{i}^{D}\tau$ [20,21]. Therefore, $P[G{}_{i}^{\prime }|G_{i}=0,A_{i}]$ and $P[G{}_{i}^{\prime }|G_{i}=1,A_{i}]$ can be defined as
(10)$$\begin{array}{c} P\left[G{}_{i}^{\prime }|G_{i}=0\right]=\left\{\begin{array}{c} 1-\lambda_{i}^{D}\tau ,\;\;\;\mathrm{if}\;G{}_{i}^{\prime }=0\\ \lambda_{i}^{D}\tau ,\;\;\;\;\;\;\;\;\mathrm{if}\;G{}_{i}^{\prime }=1\\ 0,\;\;\;\;\;\;\;\;\;\;\;\;\mathrm{otherwise} \end{array}\right. \end{array}$$
and
(11)$$\begin{array}{c} P\left[G{}_{i}^{\prime }|G_{i}=1\right]=\left\{\begin{array}{c} 1-\lambda_{i}^{C}\tau ,\;\;\;\mathrm{if}\;G{}_{i}^{\prime }=1\\ \lambda_{i}^{C}\tau ,\;\;\;\;\;\;\;\;\mathrm{if}\;G{}_{i}^{\prime }=0\\ 0,\;\;\;\;\;\;\;\;\;\;\;\;\mathrm{otherwise}. \end{array}\right. \end{array}$$

### 4.4. Cost Function

The delay to deliver the observation to the destination is used as the cost function $r(S_{i},A_{i})$. Because the delay is affected by the chosen action, we can consider the following four cases: (1) observation satellite *i* transmits its observation to the relay satellite without the compression (i.e., $A_{i}=1$); (2) observation satellite *i* transmits its observation to the ground station without the compression (i.e., $A_{i}=2$); (3) observation satellite *i* transmits its observation to the relay satellite after the compression (i.e., $A_{i}=3$); and (4) observation satellite *i* transmits its observation to the ground station after the compression (i.e., $A_{i}=4$).

For each case, the delay to deliver the observation to the destination can be calculated. For the first and third cases, because the observation is delivered through the relay satellite, we consider the latency between observation satellite *i* and the relay satellite and the latency between the relay satellite and the the ground station. In addition, the average queuing delay in the relay satellite should be considered. To sum up, the delay $L_{1}$ and $L_{3}$ to deliver the observation to the destination for the first and third cases can be obtained as $M_{i}+L_{T_{O,R}}^{W}+E\left[Q\right]L_{Q}+L_{T_{R,G}}^{W}$ and $M_{i}+L_{T_{O,R}}^{C}+E\left[Q\right]L_{Q}+L_{T_{R,G}}^{C}$, respectively, where $L_{T_{O,R}}^{W}$ ($L_{T_{R,G}}^{W}$) and $L_{T_{O,R}}^{C}$ ($L_{T_{R,G}}^{C}$) denote the latency between observation satellite *i* (relay satellite) and the relay satellite (ground station) without and with the compression, respectively. In addition, E[Q] and $L_{Q}$ are the expected queue length when the observation arrives at the relay satellite and the unit latency to handle one observation in the relay satellite, respectively. Note that the expected queue length E[Q] depends on the transmission policies of observation satellites. On the other hand, for the second and fourth cases, because the observation is directly delivered to the ground station, we only consider the latency between observation satellite *i* and the ground station. Thus, the delay $L_{2}$ and $L_{4}$ to deliver the observation to the destination for the second and fourth cases can be calculated as $M_{i}+L_{T_{O,G}}^{W}$ and $M_{i}+L_{T_{O,G}}^{C}$, respectively, where $L_{T_{O,G}}^{W}$ and $L_{T_{O,G}}^{C}$ represent the latency between observation satellite *i* and the ground station without and with the compression, respectively. Note that observation satellite *i* can directly transmit the observation to the ground station only when there is the connection between observation satellite *i* and the ground station (i.e., $G_{i}=1$). To sum up, $r(S_{i},A_{i})$ can be defined as (Equation 12).
(12)$$\begin{array}{c} r\left(S_{i},A_{i}\right)=\left\{\begin{array}{c} M_{i}+L_{T_{O,R}}^{W}+E\left[Q\right]L_{Q}+L_{T_{R,G}}^{W},\;\mathrm{if}\;A_{i}=1\\ M_{i}+L_{T_{O,G}}^{W},\;\;\;\;\;\;\;\;\;\;\;\;\;\;\;\;\;\;\;\;\;\;\;\;\;\;\mathrm{if}\;A_{i}=2\;\mathrm{and}\;G_{i}=1\\ M_{i}+L_{T_{O,R}}^{C}+E\left[Q\right]L_{Q}+L_{T_{R,G}}^{C},\;\mathrm{if}\;A_{i}=3\\ M_{i}+L_{T_{O,G}}^{C},\;\;\;\;\;\;\;\;\;\;\;\;\;\;\;\;\;\;\;\;\;\;\;\;\;\;\mathrm{if}\;A_{i}=4\;\mathrm{and}\;G_{i}=1 \end{array}\right. \end{array}$$

### 4.5. Constraint Function

To keep the usage of the computational resource of observation satellite *i* below a specific threshold, the constraint function $c_{C}\left(S_{i},A_{i}\right)$ is defined to represent whether or not to utilize the computational resource of observation satellite *i*. When observation satellite *i* compresses the observation (i.e., $A_{i}=3$ and $A_{i}=4$), it consumes the computational resource. Therefore, $c_{C}\left(S_{i},A_{i}\right)$ can be defined as
(13)$$\begin{array}{c} c_{C}\left(S_{i},A_{i}\right)=\left\{\begin{array}{c} 1,\;\mathrm{if}\;A_{i}=3\\ 1,\;\mathrm{if}\;A_{i}=4\\ 0,\;\mathrm{otherwise}. \end{array}\right. \end{array}$$

To keep the average utilization of the storage resources of observation satellite *i* below a certain level, we define the constraint function $c_{S}\left(S_{i},A_{i}\right)$ to represent whether or not to utilize the storage resources of observation satellite *i*. To delay the transmission of the observation (i.e., $A_{i}=0$), observation satellite *i* should store the observation in its storage. Thus, $c_{S}\left(S_{i},A_{i}\right)$ can be defined as
(14)$$\begin{array}{c} c_{S}\left(S_{i},A_{i}\right)=\left\{\begin{array}{c} 1,\;\mathrm{if}\;A_{i}=0\\ 0,\;\mathrm{otherwise}. \end{array}\right. \end{array}$$

### 4.6. Optimization Formulation

If a stationary multipolicy of all observation satellites π is given, a long-term average delay $\zeta_{D}$ to deliver the observation to the destination can be defined as
(15)$$\begin{array}{c} \zeta_{D}\left(\pi \right)=\lim_{T\to \infty }\frac{1}{T}\sum_{t=1}^{T}E_{\pi }\left[r\left(S^{t},A^{t}\right)\right] \end{array}$$
where $S^{t}$ and $A^{t}$ are the global state and the action at time *t*, respectively. Meanwhile, observation satellite *i* tries to keep the average utilization of its computational and storage resources below certain levels. The constraints for the average utilization of computational and storage resources, $\xi_{C}$ and $\xi_{S}$, can be represented as
(16)$$\begin{array}{c} \xi_{C}\left(\pi \right)=\lim_{T\to \infty }\frac{1}{T}\sum_{t=1}^{T}E_{\pi }\left[c_{C}\left(S^{t},A^{t}\right)\right]\leq \theta_{C} \end{array}$$
and
(17)$$\begin{array}{c} \xi_{S}\left(\pi \right)=\lim_{T\to \infty }\frac{1}{T}\sum_{t=1}^{T}E_{\pi }\left[c_{S}\left(S^{t},A^{t}\right)\right]\leq \theta_{S} \end{array}$$
where $\theta_{C}$ and $\theta_{S}$ denote the target utilization of the computational and storage resources, respectively.

Let $\pi_{i}^{*}$ and $\pi_{-i}^{*}$ denote the optimal policies (i.e., Nash equilibrium) of observation satellite *i* and all observation satellites except observation satellite *i*, respectively. Consequently, the optimal multipolicy (i.e., constrained Nash equilibrium) can be described by $\pi^{*}=\left(\pi_{i}^{*},\pi_{-i}^{*}\right)$. (When a stochastic game is constructed with a finite number of players, states, and actions, there is always Nash equilibrium (i.e., optimal solution) in the game [19].) Note that, for any other stationary policy $\pi_{i}$ of observation satellite *i*, $\zeta_{D}\left(\left(\pi_{i}^{*},\pi_{-i}^{*}\right)\right)\geq \zeta_{D}\left(\left(\pi_{i}^{},\pi_{-i}^{*}\right)\right)$ while satisfying all constraints. Meanwhile, when the policies of all observation satellites except observation satellite *i* (i.e., $\pi_{-i}$) are given, the optimal policy $\pi_{i}^{*}$ of observation satellite *i* (i.e., best-response policy) should satisfy the inequality $\zeta_{D}\left(\left(\pi_{i}^{*},\pi_{-i}^{}\right)\right)\leq \zeta_{D}\left(\left(\pi_{i}^{},\pi_{-i}^{}\right)\right)$.

Let $\phi_{i,\pi_{-i}}\left(S_{i},A_{i}\right)$ represent the stationary probability that observation satellite *i* chooses the action $A_{i}$ in local state $S_{i}$ when the stationary policies of other observation satellites $\pi_{-i}$ are given. Then, we can formulate the equivalent linear programming (LP) model and its solution $\phi_{i,\pi_{-i}}^{*}\left(S_{i},A_{i}\right)$ can be interpreted as the optimal policy of the formulated game [22,23,24]. The LP model can be formulated by
(18)$$\begin{array}{c} \min_{\phi (S,A)}\sum_{S}\sum_{A}\phi_{i,\pi_{-i}}\left(S_{i},A_{i}\right)r\left(S_{i},A_{i}\right) \end{array}$$
(19)$$\begin{array}{c} s.t.\;\;\;\sum_{S}\sum_{A}\phi_{i,\pi_{-i}}\left(S_{i},A_{i}\right)c_{C}\left(S_{i},A_{i}\right)\leq \theta_{C} \end{array}$$
(20)$$\begin{array}{c} \sum_{S}\sum_{A}\phi_{i,\pi_{-i}}\left(S_{i},A_{i}\right)c_{S}\left(S_{i},A_{i}\right)\leq \theta_{S} \end{array}$$
(21)$$\begin{array}{c} \sum_{A}\phi_{i,\pi_{-i}}\left(S_{i}^{\prime },A_{i}\right)=\sum_{S}\sum_{A}\phi_{i,\pi_{-i}}\left(S_{i},A_{i}\right)P\left[S_{i}^{\prime }|S_{i},A_{i}\right] \end{array}$$
(22)$$\begin{array}{c} \sum_{S}\sum_{A}\phi_{i,\pi_{-i}}\left({S_{i}}^{\prime },A_{i}\right)=1 \end{array}$$
(23)$$\begin{array}{c} \phi_{i,\pi_{-i}}\left({S_{i}}^{\prime },A_{i}\right)\geq 0 \end{array}$$

In order to minimize the delay to deliver the observation to the destination from observation satellite *i*, the objective function in (Equation 18) is designed. Meanwhile, the average utilization of the computational and storage resources can be maintained below the desired levels $\theta_{C}$ and $\theta_{S}$ by the constraints in (Equation 19) and (Equation 20), respectively. The constraint in (Equation 21) is for the Chapman–Kolmogorov equation derived by the chain rule for the probability and Markov property. The basic probability properties can be constrained by (Equation 22) and (Equation 23).

If any feasible solution of the LP problem exists, the stationary best-response policy of observation satellite *i* can be obtained as [25].
(24)$$\begin{array}{c} \pi_{i}^{*}\left(S_{i},A_{i}\right)=\frac{\phi_{i,\pi_{-i}}^{*}\left(S_{i},A_{i}\right)}{\sum_{A_{i}^{\prime }\in A_{i}}\phi_{i,\pi_{-i}}^{*}\left(S_{i},A_{i}^{\prime }\right)}. \end{array}$$

To obtain the optimal policies (i.e., best-response policies) of observation satellites, we develop a best-response-dynamics-based algorithm as in Algorithm 1. Each observation satellite first initializes its policy (line 1 in Algorithm 1). Then, each observation satellite transmits its current policy to other observation satellites for the interaction (line 4 in Algorithm 1). Based on the policies of other observation satellites, each observation satellite can calculate the expected queue length E[Q] when the observation arrives at the relay satellite (line 5 in Algorithm 1). Specifically, E[Q] can be calculated as $\frac{1}{2}\sum_{i}\sum_{S_{i}}\left[\phi_{i,\pi_{-i}}\left(S_{i},A_{i}=1\right)+\phi_{i,\pi_{-i}}\left(S_{i},A_{i}=3\right)\right]$. After that, each observation satellite solves the LP problem to obtain the optimal policy $\pi_{i}^{*}$ (line 6 in Algorithm 1). The algorithm is finished when the policies of all observation satellites converge (line 7 in Algorithm 1).
**Algorithm 1** Best-response-dynamics-based algorithm.1:Initialize the policies $\pi_{i}$ for ∀i.2:**repeat**3:**for** All observation satellites *i* **do**4:Transmit the policy $\pi_{i}$ to other observation satellites5:Calculate the expected queue length E[Q] when the observation is arrived at the relay satellite6:Solve the LP problem to obtain the optimal policy $\pi_{i}^{*}$7:**end for**8:**until** Stationary policies of all observation satellites converge

The complexity of solving the LP problem is relatively low [26]. For example, the complexity of Vaidya’s algorithm, which is a representative LP-solving algorithm, is a polynomial time (i.e., $O(|S_{i}|\cdot |A_{i}{|}^{3})$ for our problem). In addition, the stationary policies can be obtained within few iterations (see Section 5.1). Therefore, the proposed algorithm can be implemented without significant overhead.

## 5. Evaluation Results

For performance evaluation, we compare the proposed scheme, RNA-OTS, with the following three schemes: (1) GROUND, where the observation satellite directly transmits its observation to the ground station with/without compressing the observation; (2) RELAY, where the observation satellite always exploits the relay satellite with/without compressing the observation; and (3) RAND, where the observation satellite randomly chooses the action. The performance metrics are the average delay $\zeta_{D}$ to deliver the observation to the destination, the average utilization $\xi_{C}$ of the computational resource, and the average utilization $\xi_{S}$ of the storage resource.

The default parameter settings are as follows: The number of observation satellites is 5. The average inter-observation time $1/\lambda^{M}$ is set to [2,5] (s), where [a,b] is a random value between *a* and *b*. The average duration of the disconnection (connection) between the ground station and the observation satellite is set to [5,20]. The target utilization, $\theta_{C}$ and $\theta_{S}$, for the computational and storage resources is set 0.05 and 0.01, respectively. The latency $L_{T_{O,G}}^{W}$ ($L_{T_{O,G}}^{C}$) between the observation satellite and the ground station without the compression (with the compression) is set to 1 (0.5) (s). The latency $L_{T_{R,G}}^{W}$ ($L_{T_{R,G}}^{C}$) between the relay satellite and the ground station without the compression (with the compression) is set to 1 (0.5) (s). The latency $L_{T_{O,R}}^{W}$ ($L_{T_{O,R}}^{C}$) between the observation satellite and the relay satellite without the compression (with the compression) is set to 0.1 (0.05) (s). The unit latency $L_{Q}$ to handle one observation in the relay satellite is set to 0.01 (s).

### 5.1. Convergence to Nash Equilibrium

The process by which the policies of observation satellites are converged to the Nash equilibrium (i.e., best-response policy) is illustrated in Figure 3. From Figure 3, it can be seen that the proposed algorithm can find the Nash equilibrium within few iterations. (Note that each observation satellite randomly set its initial policy.) This indicates that our algorithm may be implemented in satellite networks without incurring significant signaling overhead.

Meanwhile, from Figure 3, it can be seen that each observation satellite decides its action with the consideration of the other observation satellites’ actions at the best response to reduce the queuing delay at the relay satellite. Specifically, in this result, after the convergence, the observation satellites 1, 2, and 3 transmit their observations to the relay satellite with relatively high probability. In this situation, if other observation satellites (i.e., satellites 4 and 5) transmit their observations to the relay satellite, the queuing delay at the relay satellite can increase significantly. To realize this fact, the observation satellites 4 and 5 maintain lower transmission probability to the relay satellite.

### 5.2. Effect of $\theta_{C}$

Figure 4 shows the effect of the target utilization of the computational resource $\theta_{C}$. From Figure 4, it can be found that RNA-OTS can achieve the lowest average delay $\zeta_{D}$ while maintaining acceptable average utilizations of computational and storage resources, $\xi_{C}$ and $\xi_{S}$ (i.e., maintaining sufficiently low average utilization of computational and storage resources). This is because observation satellites in RNA-OTS efficiently exploit given computational and storage resources and decide their transmission policies by considering other observation satellites’ policies. For example, if observation satellites in RNA-OTS are expected to have a connection with the ground station soon, observation satellites use their storage resources to store the observations. Then, after connecting to the ground station, observation satellites directly transmit the observation to the ground station. In doing so, observation satellites in RNA-OTS can effectively utilize their storage resource (i.e., the storage resource can be used only during a short time (when some observation satellites which are expected to have a connection with the ground station soon delay their transmission, the queuing delay at the relay satellite can naturally be maintained at a low level). Additionally, since observation satellites in RNA-OTS have more computational resources, they compress the observation (i.e., $A_{i}=3$ or $A_{i}=4$) to reduce the delay $\zeta_{D}$. Moreover, a certain observation satellite does not transmit its observation to the relay satellite when many other observation satellites transmit their observations to the relay satellite.

Meanwhile, from Figure 4a, it can be found that the average delay of RNA-OTS decreases with the increase in the target utilization $\theta_{C}$ of the computational resource. This is because observation satellites in RNA-OTS can aggressively compress the observation by using more computational resources under higher $\theta_{C}$, which can be observed in Figure 4b. Therefore, the observation can have a smaller size, and thus be averagely delivered within a short duration. On the other hand, other comparison schemes do not change their policies regardless of $\theta_{C}$, and therefore their average delay and average utilization of computational resources do not change (see Figure 4a,b).

### 5.3. Effect of $\theta_{S}$

Figure 5a,b show the effect of the target utilization $\theta_{S}$ for the storage resource on the average delay $\zeta_{D}$ and the average utilization of the storage resource $\xi_{S}$, respectively. As shown in Figure 5a, the average delay of RNA-OTS decreases with the increase in $\theta_{S}$. This can be explained as follows: Larger $\theta_{S}$ means that observation satellites can store their observations for longer time, which implies that observation satellites can have more chance to transmit their observation without going through the relay satellite. This can lead shorter average delay. Meanwhile, since the other comparison schemes operate with the fixed policies, their average delay and average utilization of the storage resource do not change (see Figure 5a,b). This indicates that RNA-OTS can achieve better performance gain (i.e., lower average delay) compared to the other schemes when higher $\theta_{S}$ is given. Specifically, with $\theta_{C}=0.5$, the average delay of RNA-OTS is just 13% of that of RELAY.

### 5.4. Effect of $1/\lambda_{C}$

Figure 6 shows the effect of the average duration $1/\lambda_{C}$ of the connection between the ground station and the observation satellite on the average delay $\zeta_{D}$ to deliver the observation to the destination. From Figure 6, it can be observed that the average delay of RNA-OTS decreases as $1/\lambda_{C}$ increases. This is explained in the following: A large duration $1/\lambda_{C}$ of the connection between the ground station and the observation satellite indicates that the observation satellites can transmit the observations to the ground station most of the time without conducting the delay action. In addition, because a large duration $1/\lambda_{C}$ indicates that observation satellites can transmit the observations to the ground station most of the time, the load of the relay satellite does not increase significantly (i.e., short queuing delay in the relay satellite), which leads the decreased delay $\zeta_{D}$ in delivering the observation to the destination.

### 5.5. Effect of $L_{Q}$

Figure 7 shows the effect of the unit latency $L_{Q}$ when handling one observation in the relay satellite on the average delay $\zeta_{D}$. From Figure 7, it can be observed that the average delay of all schemes except GROUND increases as $L_{Q}$ increases. This is because the queuing delay at the relay satellite increases as $L_{Q}$ increases. However, because the observation satellites in RNA-OTS maintain a small length of queue in the relay satellite by considering the other observation satellites’ actions, its increasing rate is smallest. Note that, because GROUND does not use the relay satellite, the average delay $\zeta_{D}$ is constant regardless of $L_{Q}$.

## 6. Conclusions

This paper proposes a resource- and neighbor-aware observation transmission scheme (RNA-OTS) where each observation satellite periodically decides whether or not to exploit its own resource and the resource of the relay satellite. For the optimal decision of each observation satellite in a distributed manner, the operation of observation satellites is modeled by means of a constrained stochastic game, and a best-response-dynamics-based algorithm is devised to find the Nash equilibrium. The evaluation results demonstrate that RNA-OTS can decrease the delay to deliver the observation to the destination by up to 87% compared to a relay-satellite-based scheme while providing a sufficient low average utilization of computational and storage resources. Moreover, it can be observed that RNA-OTS optimizes its transmission policy by considering its operating environment (e.g., connection duration between the ground station and the observation satellite). In our future work, we will extend the proposed scheme in order to learn the optimal policy without prior information on the operational environment. In addition, we will conduct the route optimization of the observation satellites to further reduce the observation delivery delay.

## Figures and Tables

**Figure 1 sensors-23-04889-f001:**
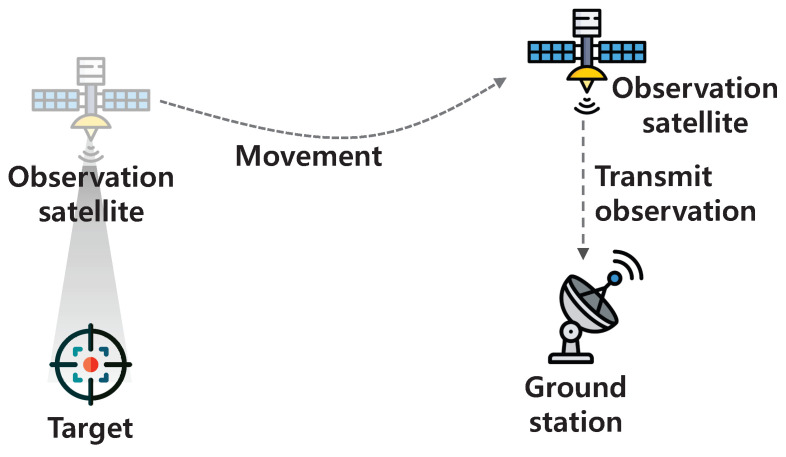
Motivating example.

**Figure 2 sensors-23-04889-f002:**
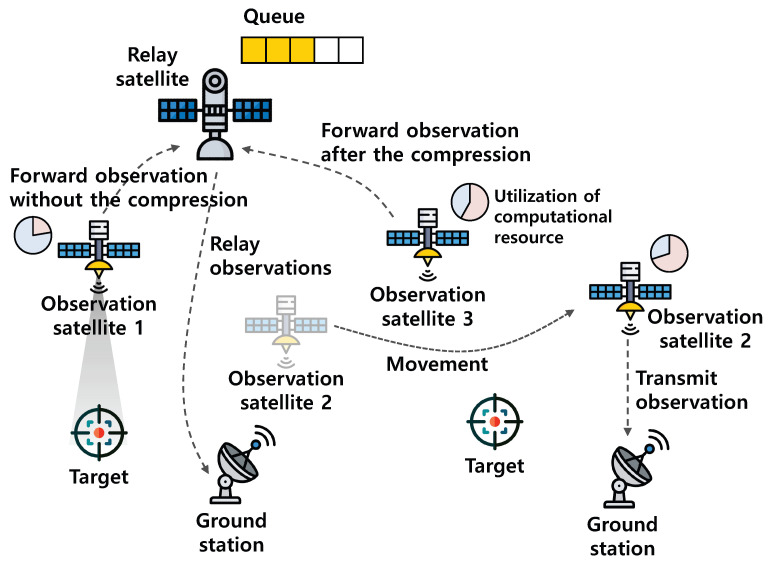
System model.

**Figure 3 sensors-23-04889-f003:**
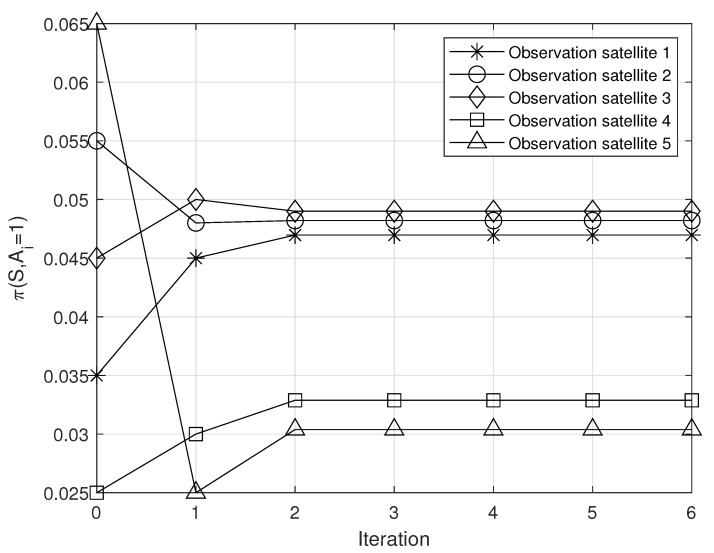
Convergence to optimal policies.

**Figure 4 sensors-23-04889-f004:**
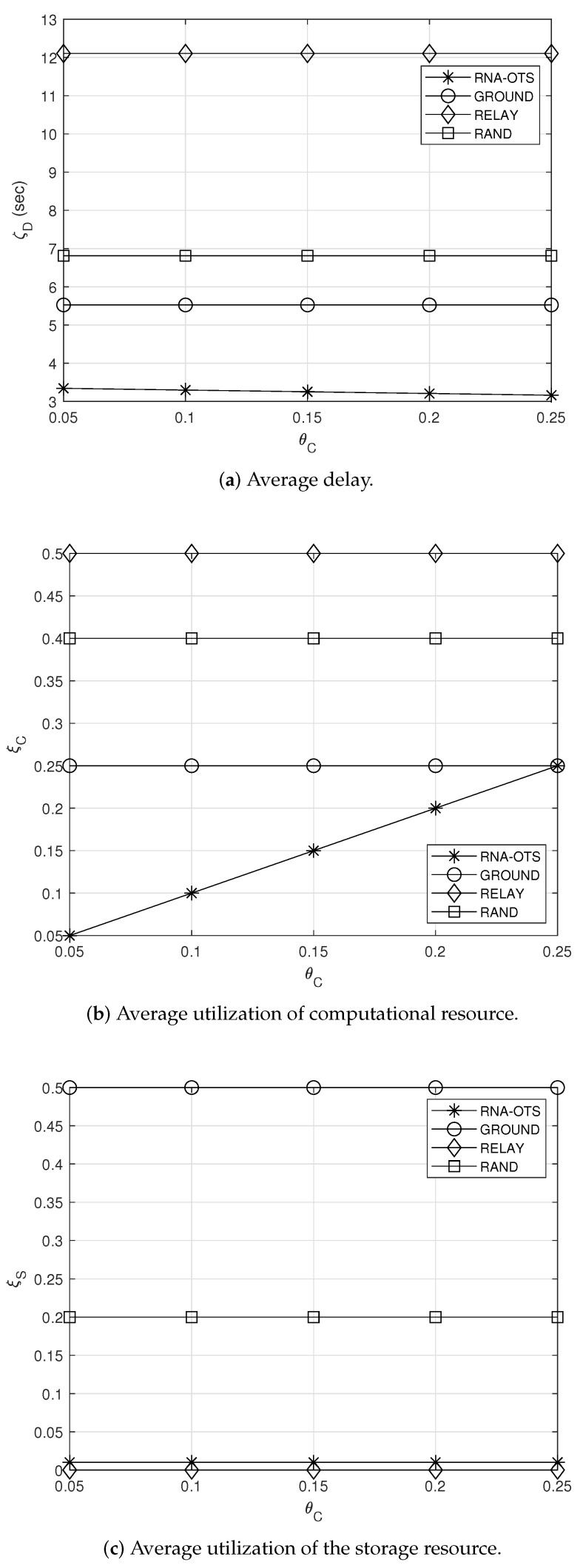
Effect of the target utilization of the computational resource.

**Figure 5 sensors-23-04889-f005:**
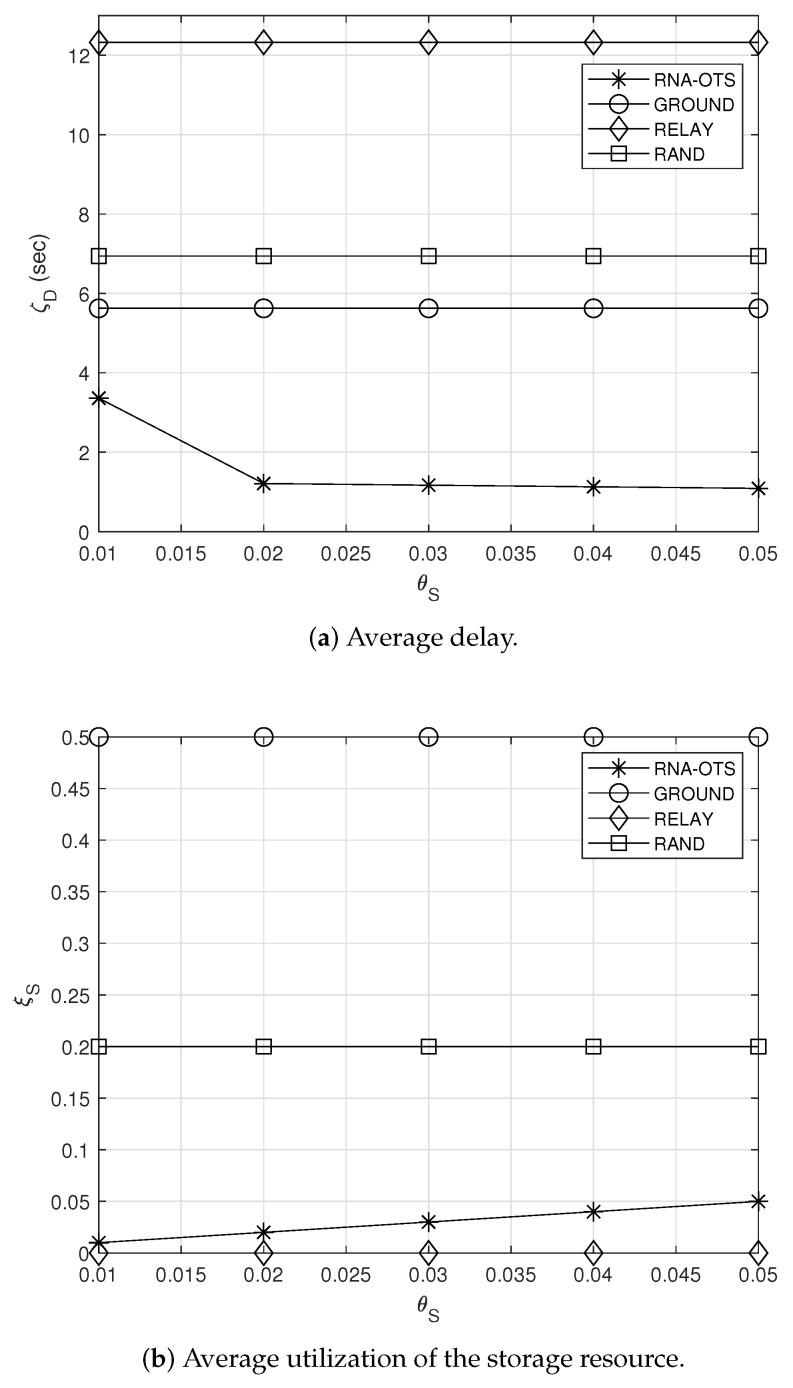
Effect of the target utilization of the storage resource.

**Figure 6 sensors-23-04889-f006:**
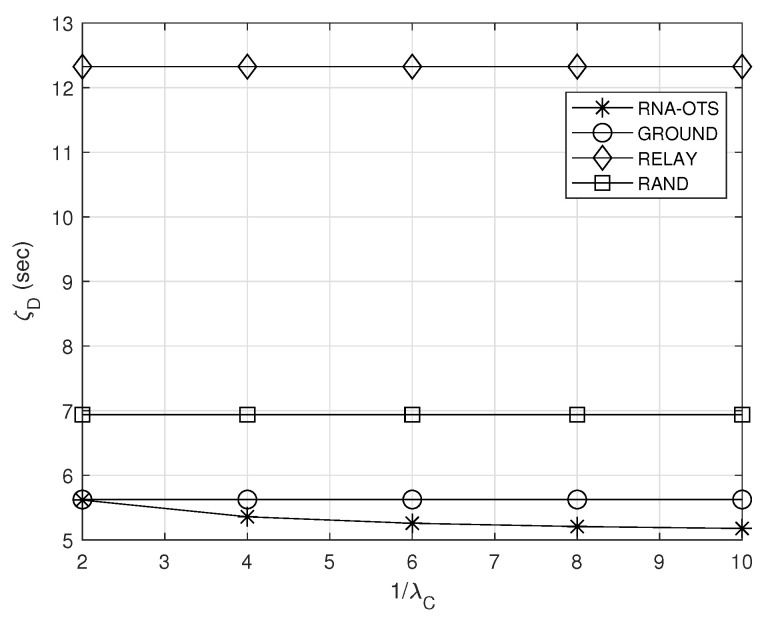
Effect of the average duration of the connection between the ground station and the observation satellite on the average delay.

**Figure 7 sensors-23-04889-f007:**
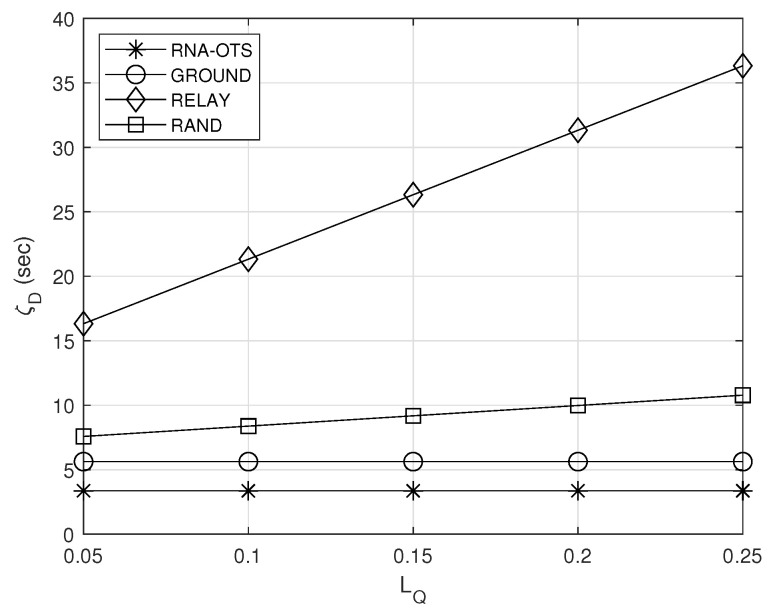
Effect of the unit latency when handling one observation in the relay satellite on the average delay.

**Table 1 sensors-23-04889-t001:** Comparison of collaborative approaches.

Ref	Objective	Decision Element	Method
[13]	Maximize the throughput	Transmission schedule	Algorithm
[14]	Maximize the throughput	Transmission schedule	MDP
[15]	Maximize the number of completed tasks & minimize the total power consumption	Task schedule	Algorithm
[16]	Maximize the number of completed tasks	Task schedule	Algorithm
[17]	Maximize the task completion rate	Task schedule	Algorithm

**Table 2 sensors-23-04889-t002:** Summary of notations.

Notation	Description
$S_{i}$	Local state space of observation satellite *i*
S	Global state space
$M_{i}$	Observation state space
$G_{i}$	Ground station state space
$A_{i}$	Local action space of observation satellite *i*
A	Global action space
$L_{T_{O,R}}^{W}$	Latency between the observation satellite and the relay satellite without the compression
$L_{T_{O,R}}^{C}$	Latency between the observation satellite and the relay satellite with the compression
$L_{T_{R,G}}^{W}$	Latency between the relay satellite and the ground station without the compression
$L_{T_{R,G}}^{C}$	Latency between the relay satellite and the ground station with the compression
$L_{T_{O,G}}^{W}$	Latency between the observation satellite and the ground station without the compression
$L_{T_{O,G}}^{C}$	Latency between the observation satellite and the ground station with the compression
E[Q]	Expected queue length when the observation arrives at the relay satellite
$L_{Q}$	The unit latency to handle one observation in the relay satellite
$\zeta_{D}$	Average delay to deliver the observation to the destination
$\xi_{C}$	Average utilization of the computational resource
$\xi_{S}$	Average utilization of the storage resource
$\theta_{C}$	Target utilization of the computational resource
$\theta_{S}$	Target utilization of the storage resource

## Data Availability

Not applicable.

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
