# Peer review of "Resource- and Neighbor-Aware Observation Transmission Scheme in Satellite Networks"

_sensors, 2023, doi:10.3390/s23104889_

Round 1

Reviewer 1 Report

The paper proposes a resource and neighbor-aware observation transmission scheme for observation satellites in satellite networks, which allows each observation satellite to decide whether to use its resources and the resources of the relay satellite at each time epoch based on its resource utilization and transmission policies of neighbor observation satellites. The proposed scheme is modeled as a constrained stochastic game and a best response dynamics-based algorithm is used to find Nash equilibrium. The evaluation results show that the proposed scheme can significantly reduce the transmission delay while ensuring low average utilization of resources of the observation satellite.

Although the manuscript is well-written, there are a few areas where additional improvements could be made to strengthen the work. Firstly, adding a comparison table at the end of the related work section to compare the proposed scheme with existing schemes would help readers understand the advantages of the proposed scheme over other approaches. Additionally, including a figure in the introduction section could provide a visual aid to help readers better understand the problem being addressed.

Furthermore, the future work section in this paper is limited, and it would be beneficial to expand on this section to provide additional insights and avenues for further research.
Finally, in section 5.1. Convergence to Nash equilibrium, the sentence Ther observation satellites 4 and 5 transmit their observations to the 275 ground station with lower probability to reduce the queuing delay at the relay satellite" could benefit from further clarification to ensure that readers understand the context and rationale behind this decision.

Author Response

We really appreciate your time and valuable comments. We have carefully revised our paper taking into consideration of your comments and suggestions. The attached file has detailed responses to your comments (C: Comment, A: Answer). Please note that modifications are marked as “bold” in the revised manuscript.

Reviewer 2 Report

Moderate editing of English language is needed.

Author Response

(The authors gave the same response as above.)

Reviewer 3 Report

Article is very interesting and devoted to resource optimization in Satellite Networks. But there some improvements that can be made.

1. Novelty of proposed scheme is not clear (mainly in Abstract) - neighbor-aware observation transmission approach (OTS) is new or already known?

2. Stochastic game theory and algorithms to find Nash equilibrium seem to be well known. More reference on this subject can be made in the Introduction.

3. The comparison is made with relay satellite scheme, but not with known neighbor-aware observation scheme. The paper contains references no later than 2020, some more fresh references can be added to compare with proposed approach. 

4. The description of computational resources calculation can be improved for Figure 4. The gain of proposed scheme is not clear from the simulation results, the presentation should be improved. RNA-OTS can decrease the delay to deliver the observation to the destination up to 337

5. In the Conclusion Authors say: "RNA-OTS can decrease the delay to deliver the observation to the destination up to 87% compared to a relay satellite-based scheme" - this is not clear form the simulation results. Also the phrase "sufficient low average utilization of computational and storage resources" is needed to be more clearly justified on the basis of simulation results. Also the dependence of observed satellites number on the simulation results can be added for the proposed scheme comparison.

Wikipedia references (18 and 22) can be be replaced by references to more serious literature (books, articles). Format of Figure 4 (size) is not as convenient as for Figure 5. Inscriptions of coordinate axes must contain units of physical parameters that are presented on the curves.

in my opinion, the English language of the article is quite understandable and may need only minor changes.

Author Response

(The authors gave the same response as above.)

Round 2

Reviewer 2 Report

The authors have basically addressed my concerns, no further comments.

Minor editing of English language is required

Author Response

We really appreciate your time and valuable comments. We have carefully revised our paper taking into consideration of your comments and suggestions. We have added the response letter to reviewers based on your comments. Please note that modifications are marked as “bold” in the revised manuscript. 

Reviewer 3 Report

Thank You for Your work. The comments were taken into account and the article was improved.

Maybe minor corrections are needed.

Author Response

(The authors gave the same response as above.)
